# Pepper-Mediated Green Synthesis of Selenium and Tellurium Nanoparticles with Antibacterial and Anticancer Potential

**DOI:** 10.3390/jfb14010024

**Published:** 2022-12-31

**Authors:** Veer Shah, David Medina-Cruz, Ada Vernet-Crua, Linh B. Truong, Eduardo Sotelo, Ebrahim Mostafavi, María Ujué González, José Miguel García-Martín, Jorge L. Cholula-Díaz, Thomas J. Webster

**Affiliations:** 1Department of Chemical Engineering, Northeastern University, Boston, MA 02115, USA; 2Nanomedicine Science and Technology Center, Northeastern University, Boston, MA 02115, USA; 3School of Engineering and Sciences, Tecnologico de Monterrey, Monterrey 64849, Mexico; 4Stanford Cardiovascular Institute, Stanford University School of Medicine, Stanford, CA 94305, USA; 5Department of Medicine, Stanford University School of Medicine, Stanford, CA 94305, USA; 6Instituto de Micro y Nanotecnología, IMN-CNM, CSIC (CEI UAM+CSIC), Isaac Newton 8, 28760 Tres Cantos, Spain; 7School of Health Sciences and Biomedical Engineering, Hebei University of Technology, Tianjin 300401, China; 8School of Engineering, Saveetha University, Chennai 602105, India; 9Department of Materials, Federal University of Piaui, Teresina 64001, Brazil

**Keywords:** nanoparticles, peppers, chalcogens, biomedicine, antibacterial, anticancer

## Abstract

The production of nanoparticles for biomedical applications (namely with antimicrobial and anticancer properties) has been significantly hampered using traditional physicochemical approaches, which often produce nanostructures with poor biocompatibility properties requiring post-synthesis functionalization to implement features that such biomedical applications require. As an alternative, green nanotechnology and the synthesis of environmentally friendly nanomaterials have been gaining attention over the last few decades, using living organisms or biomolecules derived from them, as the main raw materials to produce cost-effective, environmentally friendly, and ready-to-be-used nanomaterials. In this article and building upon previous knowledge, we have designed and implemented the synthesis of selenium and tellurium nanoparticles using extracts from fresh jalapeño and habanero peppers. After characterization, in this study, the nanoparticles were tested for both their antimicrobial and anticancer features against isolates of antibiotic-resistant bacterial strains and skin cancer cell lines, respectively. The nanosystems produced nanoparticles via a fast, eco-friendly, and cost-effective method showing different antimicrobial profiles between elements. While selenium nanoparticles lacked an antimicrobial effect at the concentrations tested, those made of tellurium produced a significant antibacterial effect even at the lowest concentration tested. These effects were correlated when the nanoparticles were tested for their cytocompatibility and anticancer properties. While selenium nanoparticles were biocompatible and had a dose-dependent anticancer effect, tellurium-based nanoparticles lacked such biocompatibility while exerting a powerful anti-cancer effect. Further, this study demonstrated a suitable mechanism of action for killing bacteria and cancer cells involving reactive oxygen species (ROS) generation. In summary, this study introduces a new green nanomedicine synthesis approach to create novel selenium and tellurium nanoparticles with attractive properties for numerous biomedical applications.

## 1. Introduction

Nanotechnology is primarily concerned with the synthesis of different nanomaterials of variable sizes, shapes, chemical compositions, and controlled dispersity with potential use in human applications. In the case of medicine, it has given rise to the field of nanomedicine [1,2]. Traditionally, chemical and physical methods have been extensively employed to produce valuable nanostructures, including, but not limited to solvothermal-based, chemical reduction, and laser ablation methods [3,4]. Although these traditional approaches successfully produce pure and well-defined nanoparticles (NPs), they often trigger the release of toxic by-products from the newly synthesized NPs or their reaction mixtures, employ hard reaction conditions and toxic solvents, fail to offer good cost-effectiveness production, and/or pose threats to the environment [5,6]. Therefore, research has been shifting away from these traditional approaches and moving towards more responsible and eco-friendly solutions that can achieve comparable results in terms of nanoparticle properties. 

One of these alternatives comes from green nanotechnology, defined as the use of living organisms and natural raw materials to produce NPs [7,8]. Among all the potential protocols available for the greener production of NPs, plant extracts are becoming increasingly valuable due to their wide range of available raw materials, strong therapeutic potential, and favorable balance between the cost-effectiveness and scalability that they provide [9,10,11]. Plant extracts, once isolated from their raw materials, often act both as reducing and stabilizing agents for the synthesis of NPs, and strongly influence the size, morphology, and properties of NPs via the chemical characteristics of the active compounds within the plant extracts. These active compounds range from terpenoids, polyphenols, sugars, and alkaloids, to phenolic acids and proteins whose content and concentration depend on the plant extract being used, all playing an important role in the bioreduction of metal ions [12,13].

From the nanomaterial synthesis point of view, peppers can be used as one of the raw materials to produce biologically relevant plant extracts. Pepper extracts are rich in alkaloids peptides, which contain mostly basic nitrogen atoms, polysaccharides, amino acids, and vitamins, all of which can reduce metal ions to NPs and subsequently help in their stabilization. To date, numerous NPs have been prepared using pepper extracts, mainly silver NPs (AgNPs) [14,15,16,17,18,19], zinc oxide NPs (ZnONPs) [20], iron oxide-palladium nanocomposites [21], gold NPs (AuNPs) [22] or palladium NPs (PdNPs) [23]. Most of the time, both black or green pepper extracts have been used as reducing and capping agents, producing NPs with biomedical and environmental applications. 

Among the different NP systems to be used, selenium (Se) and tellurium (Te) NPs were chosen as the focus of this work, since both chalcogen elements are relevant as biomedical agents and as emerging materials with attractive biomedical applications at the nanoscale, respectively. Se is a relevant micronutrient whose nanoscale form has attracted plenty of attention due to its biocompatibility, bioavailability, and low toxicity [24,25]. Moreover, SeNPs can act as relevant antimicrobial, antioxidant and chemo-preventive agents that aid in the detoxification of heavy metal exposure and the prevention of DNA oxidation [26,27]. On the other hand, and although widely less understood and used in terms of its biomedical potential, nanoscale Te offers an interesting niche for the development of medically relevant nanotherapeutics [28]. Despite the high toxicity of Te oxyanions to the biota [29], its nanoparticle form shows a significantly reduced cytotoxic profile which has allowed researchers to find some important applications as antimicrobial and anticancer agents, as presented by our group and others in the literature [30,31,32,33,34,35]. 

Se- and TeNPs have been synthesized using different green nanotechnology approaches [28,36], including plant extracts. However, to our knowledge, this report is the first time that pepper extracts are used for the synthesis of both types of NPs. In this paper, Jalapeño (*Capsicum annuum*) and habanero (*Capsicum chinese*) were selected as both reducing and stabilizing agents for NP synthesis due to an extreme abundance of useful plant metabolites and active raw materials. Both pepper extracts were isolated and used for the synthesis of NPs, which were purified and characterized for their physicochemical properties. Then, the nanosystems were tested for their antibacterial properties against gram-negative and gram-positive bacterial isolates, as well as for their cytotoxic profiles with healthy dermal fibroblasts and cancerous human melanoma cells, revealing insights into the mechanisms of cell death associated to reactive oxygen species production. The main goal of the work was to establish the use of pepper extracts as a reproducible, environmentally friendly, and cost-effective method to produce valuable plant-mediated chalcogen NPs with potential for biomedical applications. 

## 2. Materials and Methods

### 2.1. Se- and TeNPs Synthesis

The Se- and TeNP synthesis methods were based on the use of only two raw materials (pepper extracts and selected salts of Se and Te) as well as an easily available energy source (microwaves) to produce localized heating within a mixture of the raw materials. Briefly, equally sized Jalapeño and Habanero peppers (acquired from a local supermarket) were thoroughly rinsed with warm water, ethanol, and cold water, in a consequential order, with the aim to remove any contaminants and/or debris that could interfere with the reaction. Then, 80 g of each one of the peppers were weighed, finely chopped into small pieces (roughly 1 × 1 cm^2^) and placed into 300 mL of deionized (DI) water and heated under 80 °C for 30 min to release most of the water-soluble phytocomponents to prepare the extract. After 30 min, the solids were removed, and the liquid was filtered using a 0.22 µm membrane filter (Millipore), transferred to different vials, and thereafter stored at 4 °C in the dark for further experiments. The extracts were labeled as JP and HB when isolated from Jalapeño and Habanero peppers, respectively. 

For the preparation of the NPs, sodium selenite (Na_2_SeO_3_, ThermoFisher, Walthan, MA USA) and sodium tellurite (Na_2_TeO_3_, ThermoFisher, Walthan, MA, USA) were used as the salts to be reduced into NPs. Briefly, 20 mM solutions were prepared in DI-water and mixed with the respective pepper extracts to a final concentration of 10 mM. The mixture was placed inside a microwave oven (Argos Technologies, Vernon Hills, IL, USA) and heated via a short 30 s cycle at 120 V, 170-Watt microwave, heating at a range between 90 and 100 Celsius for 5–10 s (repeated cycles, three in total, with a 20 s cooling in between). After this heating process, the solution was cooled down for several minutes to room temperature. The final solution, containing the NPs, was centrifuged at 10,000 rpm for 30 min and washed three times in DI-water to remove any unreacted materials. Then, the NPs in the form of a fine pellet were reconstituted in DI-water and freeze-dried overnight (Labconco Freezone 4.5 Liter Freeze Dry System, Kansas City, MO, USA), providing a powder that was weighed and resuspended in DI-water at a concentration of 1 mg/mL.

### 2.2. Physical and Chemical Characterization of the NPs

#### 2.2.1. UV-Visible Spectroscopy

UV-visible spectroscopic characterization was used to follow the progress of the synthesis of the NPs and the production variability within the extracts in a SpectraMax M3 spectrophotometer (Molecular Devices, Sunnyvale, CA, USA). The microwave heating method was compared with the traditional water-bath heating method (in which the Se and Te salts and the extracts were mixed and heated on a hot plate with stirring until color changes comparable to the ones found in the microwave synthesis were achieved). A 96-well plate Falcon clear plate was prepared with different dilutions of the extracts, the metal salt solutions, and the mixture of salts and extracts, and then a full absorbance spectrum was recorded from 200 to 800 nm with 10 nm spacing.

#### 2.2.2. Fourier-Transform Infrared Spectroscopy

Structural analyses of the Se- and TeNPs were conducted by Fourier-transform infrared (FT-IR) spectroscopy using an FT-IR spectrophotometer, PerkinElmer Spectrum 400 FT-IR/FT-NIR in attenuated total reflectance (ATR) mode. For FT-IR spectroscopy measurements, 5 μg of each dried sample was used. The FT-IR spectra were scanned in the range of 500 to 4000 cm^−1^ with a resolution of 4 cm^−1^. The Spectrum™ 10 STD software (PerkinElmer, Waltham, MA, USA) was used for spectra normalization and baseline correction.

#### 2.2.3. High-Resolution Transmission Electron Microscopy Analysis

Thorough morphological characterization of the NPs was accomplished using transmission electron microscopy (TEM) (JEM-1010 TEM; JEOL Inc., Peabody, MA, USA). To prepare the samples for imaging, the NPs were dried on 300-mesh copper-coated carbon grids (Electron Microscopy Sciences, Hatfield, PA, USA). 

#### 2.2.4. Scanning Electron Microscopy

SEM images were taken in an FEI Verios 460 scanning electron microscope. For observation, 10 µL of the Jalapeño or Habanero pepper synthesized Se- or TeNPs solutions were deposited on clean Si substrates and allowed to dry for 24 h. SEM imaging conditions included a 2 kV acceleration voltage and a 13 pA electron beam current. For EDX characterization, an EDAX detector coupled to an electron microscope was used. SEM conditions here involved a 400 pA current and different acceleration voltages (5/10/20 kV) for the different samples as specified in the discussion section of Section A.3 in Appendix A.

#### 2.2.5. X-ray Diffraction Analysis

Powder XRD patterns were obtained with a Rigaku MiniFlex 600 operating with a voltage of 40 kV, a current of 15 mA, and Cu-K_α_ radiation (λ = 1.542 Å). All XRD patterns were recorded at room temperature with a step width of 0.01 (2Θ) and scan speed of 10 °/min. Samples for XRD analysis were prepared by drying 2 mL of the colloids on the sample holder.

#### 2.2.6. Stability Analysis

To analyze the stability of the samples, Zeta-potential measurements were completed in fresh and 60-day-old nanoparticles using a Zetasizer NANO ZSP (Malvern Panalytical, Cambridge, UK). The nanoparticles that were kept for 60 days were stored in aqueous media after purification (as stated above). No further processing was conducted on the structures. 

### 2.3. Testing the NPs for Biomedical Applications

#### 2.3.1. Preparation of the Bacterial Cultures and Testing Antimicrobial Efficacy

Multidrug-resistant Escherichia coli (MDR E. coli) (ATCC BAA-2471; ATCC, Manassas, VA, USA) and Methicillin-resistant Staphylococcus aureus (MRSA) (ATCC 4330; ATCC, Manassas, VA, USA) bacterial strains were employed to test the NPs’ antibacterial efficacy. The bacterial cultures on agar plates were kept at 4 °C prior to use. After being diluted in 5 mL of a sterile Luria–Bertani (LB) (bioPLUS, bioWORLD, Dublin, OH, USA) medium in a 15-mL Falcon centrifuge tube, a single colony from the agar plates was grown for 24 h at 37 °C and 200 RPM. Using a spectrophotometer, the optical density (OD) of the bacterial cultures was determined at 600 nm (SpectraMax M3, Molecular Devices, Sunnyvale, CA, USA). Following that, the bacterial suspensions were diluted to a concentration of 10^6^ colony-forming units per milliliter (CFU/mL) (information gathered using standard curves of OD600) and kept at 4 °C until needed. Utilizing a plate reader equipped with a SpectraMax® Paradigm® Multi-Mode Detection Platform (Molecular Devices, Silicon Valley, CA, USA), growth curves and other bacterial tests were carried out. 

For antimicrobial experiments, a fixed volume of each NP solution was mixed with a fixed volume of bacterial colonies in the LB medium and added at various doses. The mix was then added to a 96-well plate (Thermo Fisher Scientific, Waltham, MA, USA); 100 µL of bacteria were combined with 100 uL of LB medium devoid of NPs for the untreated controls. Each well’s ultimate capacity was 200 µL. After the plate was set up, no shaking was used for 24 h as the absorbance of all samples was measured at 600 nm on a plate reader every two minutes. The absorbance brought on by the NPs was measured using negative controls made up exclusively of medium and NPs. 

Assays for colony counts were also carried out. In a 96-well plate, bacteria were seeded and exposed to various NP concentrations for 8 h at 37 °C in an incubator. The 96-well plate was then taken out of the incubator, and all the samples were diluted with phosphate buffer saline (PBS) in a series of vials to either 100, 1000, or 10,000 times their original concentration. Then, three drops of each dilution from a 10 µL aliquot were added to an LB-Agar plate and left to incubate for 8 hours at 37 °C. At the conclusion of the incubation period, the number of colonies that had developed in each plate was counted.

#### 2.3.2. Testing the Effect of the Nanomaterials towards Human Cells

Primary human dermal fibroblasts (HDF) from Lonza (CC-2509, AMP, Hopkinton, Massachusetts, USA), and melanoma cells from ATCC (CRL-1619, Manassas, VA, USA), were cultured at 37 °C and 5% CO_2_ in a humidified environment in Dulbecco’s Modified Eagle Medium (DMEM; Thermo Fisher Scientific, Waltham, Massachusetts, USA), supplemented with 10% fetal bovine serum (Thermo Fisher Scientific). The cells were trypsinized and placed onto 96-well tissue culture plates when they were 70% confluent (as measured by microscopy and cell counters) (Thermo Fisher Scientific). A final seeding density of 5x10^4^ HDF and melanoma cells per well in 100 µL of cell media was used. For the metabolic and cell viability experiments that are detailed below, the seeded well plates were maintained in a humidified environment at 37 °C and 5% CO_2_.

To determine the cellular metabolic activity and assess the cytotoxicity, (3-(4,5-Dimethylthiazol-2-yl)-5-(3-carboxymethoxyphenyl)-2-(4-sulfophenyl)-2H tetrazolium) (MTS) assays (CellTiter 96^®^ Aqueous One Solution Cell Proliferation Assay, Promega, Madison, WI, USA) were used. Cells were cultured for 24 h at 37 °C with 5% CO_2_ in a humidified incubator. The culture media was then changed for 100 µL of new cell medium that included Se- and TeNPs at varying concentrations ranging from 25 to 175 µg/mL. NPs underwent 30 min of UV sterilization prior to in vitro testing. Cells were grown for a further 24 h under identical circumstances, after which they were washed with PBS and the medium was replaced with 100 µL of the MTS solution (prepared using a mixing ratio of 1:5 of MTS:medium). The 96-well plate was incubated for 4 h after the solution was added to allow for a color change. Then, the absorbance at 490 nm was measured using an absorbance plate reader (SpectraMAX M3, Molecular Devices, San Jose, CA, USA) to determine if the cells were still viable after being exposed to NPs. Following the manufacturer’s instructions, the average absorbance of each sample was divided by the average absorbance of the control sample and then multiplied by 100 to determine the cell viability. The 96-well plate also had controls that either contained cells and media or simply media to gauge the medium’s absorbance and gauge the typical proliferation of cells devoid of nanoparticles. Two separate cell experiments—one lasting just 24 h and the other 48 h—were conducted.

#### 2.3.3. Cell Fixation and SEM Imaging (For Bacteria and Human Cells)

Both bacterial strains (MDR E. coli and MRSA) were inoculated into 5 mL of a sterile LB medium in a 50 mL Falcon conical centrifuge tube and incubated at 37 °C/200 rpm for 24 h to determine the presence of bacterial cells. Then, a spectrophotometer was used to detect the optical density at 600 nm (OD600). Before measuring the optical density, the overnight suspension was diluted to a final bacterial concentration of 10^6^ colony-forming units per milliliter (CFU/mL). In a 6-well plate with a glass coverslip affixed to the bottom, the LB medium and bacterial solution were combined with a chosen 75 µg/mL concentration of Se- and TeNPs. Before the experiment, the coverslips were pre-treated with poly-lysine to improve cell adherence. The plate spent 8 h at 37 °C inside an incubator.

The cells were seeded in a 6-well plate with a glass coverslip (Fisher Brand) glued to the bottom to identify the primary human skin fibroblasts and melanoma cells. The medium was removed and replaced with a new one having a concentration of 50 µg/mL of SeNPs and TeNPs during a 24-h period of incubation at 37 °C in a humidified incubator with 5% carbon dioxide (CO_2_). A further 24 hours of cell culture took place under identical circumstances. After the tests, the coverslips were treated for an hour with the main fixative solution containing 0.1 M sodium cacodylate buffer solution and 2.5% glutaraldehyde. The coverslips were then rinsed three times for 10 minutes with a 0.1 M sodium cacodylate buffer in place of the fixative solution. A 1% solution of osmium tetroxide (OsO_4_) in a buffer was used for post-fixation, which took place for an hour. The coverslips were then rinsed three times with buffer and gradually dehydrated using 35, 50, 70, 80, 95, and 100% ethanol—three times for the latter. The coverslips were then dried using a Samdri®-PVT-3D Critical Point drier and liquid CO_2_-ethanol exchange. After being treated with liquid graphite, the coverslips were placed on SEM stubs using carbon adhesive tabs from Electron Microscopy Sciences (EMS), and then a Cressington 208HR High Resolution Sputter Coater (Cressington, Liverpool, United Kingdom) was used to sputter coat them with a thin coating of platinum. Using an SEM, digital pictures of the treated and untreated microorganisms were captured (Hitachi S-4800, Chiyoda City, Tokyo, Japan).

#### 2.3.4. Mechanistic Analysis of the Se- and TeNP Interaction with Cells

To assess intracellular reactive oxygen species (ROS) in melanoma cells, the ROS detection assay kit was used with 2′,7′-dichlorodihydrofluorescein diacetate (ab113851, DCFDA Cellular ROS Detection Assay Kit, Abcam, Waltham, MA, USA). A fluorescent probe called DCFDA was used to quantify ROS such as peroxyl radicals, hydrogen peroxide, and peroxynitrite. Melanoma cells were cultured for a brief amount of time in 100 µL of the working media (DMEM media with 10% FBS and% P/S (penicillin-streptomycin)) in 96-well plates at a cell density of 2.5 × 10^4^ cells per well. At 485 nm excitation and 535 nm emission wavelengths, ROS was found after 24 h of incubation at 37 °C in a 5% CO_2_-humidified environment. A TBHP (Tert-Butyl Hydrogen Peroxide) solution was added to three wells at a final concentration of 500 µM as a positive control. To reach the final concentration of 25 µM DCFDA, 100 µL of 50 µM DCFDA in the working media was added to each well 45 minutes before the incubation was finished. A fluorescence microplate reader operating at 485 nm excitation and 535 nm emission wavelengths was used to measure the end-point fluorescence.

### 2.4. Statistical Analysis

To verify the accuracy of the results, all tests were conducted three times (N = 3) unless otherwise stated. Student’s t-tests were used to determine statistical significance; an alpha value of 0.05 or less was considered statistically significant. The mean and standard deviation are reported.

## 3. Results and Discussion

### 3.1. Se- and TeNP Synthesis and Purification

The environmentally friendly synthesis of Jalapeño (JP)- and Habanero (HB) -mediated Se^−^ and TeNPs was carried out by a reduction of their respective oxyanions, SeO_3_^2−^ and TeO_3_^2−^. Both plant extracts are rich in phytocomponents that acted as both reducing and capping agents with the assistance of a microwave heating source. Details of the synthesis process in terms of coloration changes with time can be found in Appendix A (Section A.1). A schematic representation of the entire synthesis and purification process can be found in Figure 1. 

The reduction of Se and Te ions to NPs can be explained by the composition of the extracts. Both JP and HB extracts contain several biomolecules, including, but not limited to alkaloids, flavonoids, proteins, polysaccharides, amino acids and vitamins [37]. These biomolecules, when present in an aqueous solution, act as reducing agents that react with metal ions and often become scaffolds or templates to direct the formation and growth of NPs in the solution [38]. The mechanism responsible for reduction is associated with the agglomeration of small elemental nuclei in a process known as Ostwald ripening [39], which can be conceptualized in the following steps: (1) the ions are trapped on the surface of the predominant biomolecules present in the extract via electrostatic interactions; (2) the ions are then reduced by these biomolecules (mostly polysaccharides and proteins) leading to changes in their conformation and structure, as well as the formation of small elemental nuclei and (3) the nuclei subsequently grow by further reduction of ions and due to the accumulation of the nuclei, leading to larger NPs, whose growth is quenched by the capping of the same biomolecules associated with the reduction reaction or others also present in the solution. In the case of the Jalapeño and Habanero extracts, most of the molecules associated with the production of NPs are flavones, flavonols, thiamine and capsaicin [16,19]. Other molecules that are rich in reducing amine groups might have been also involved [14]. 

### 3.2. UV-Visible Spectroscopic Characterization of the Samples

UV-visible spectroscopic analysis showed the formation of Se- and TeNPs when using different concentrations of Na_2_SeO_3_ (Figure 2A,C) and Na_2_TeO_3_ (Figure 2B,D), respectively. The JP and HB extract absorbance values were subtracted to only show the contribution of Se and Te to the reaction. In the case of the Se-based samples (HB- and JP-SeNPs), a clear absorption band around 380 nm was observed in the colloids starting at a Se precursor concentration of 1 mM. This optical signal can be correlated with the formation of SeNPs with particle sizes in the range of 90–110 nm [40]. The absorption band had a significant increase at a concentration of 5 mM Na_2_SeO_3_ for both JP- and HB-experiments (Figure 2A,C), with a slight further increase for the 10 mM concentration. For the Te-based nanosystems, two absorption bands could be observed at around 300 and 380 nm when a 1 mM Te-precursor (and higher concentration) was used. The signal at 300 nm is due to the direct transition from the valence band (p-bonding triplet) to the conduction band (p-antibonding triplet), whereas the second signal may be assigned to a forbidden direction transition [41]. Moreover, intense absorption bands can be observed when the system concentration increased to a 10 mM concentration. The experimental evidence showed that a higher content of Se and Te precursors in solution led to an apparent increase in the production of nanoparticles, with more significant differences between 5 and 10 mM for the Te-based experiments.

### 3.3. FT-IR Spectroscopy Characterization of the Samples

The FT-IR spectra of HB-SeNPs, HB-TeNPs, JP-SeNPs and JP-TeNPs are shown in Figure 3. In general, the samples showed a consistently strong and broad signal around 3279 cm^−1,^ which is characteristic of the stretching OH/NH groups found in the capsaicin molecule [42]. At 2933 cm^−1^, there is a signal that is representative of the C-H stretching vibration [40]. The band around 1650 cm^−1^ is assigned to the stretching vibration of amide I C=O groups present in capsaicin and proteins [43,44], while the band at around 1590 cm^−1^ is representative of the stretching vibrations of amide II and amide III bands [42]. The CH_2_ vibrational modes can be observed with a medium intensity from 1407 to 1350 cm^−1^ [42]. The ether (C-O-C) bonds, C-N and C-O groups found in capsaicin can be assigned to the signals around 1100–935 cm^−1^ [42]. Thus, we infer that the nanoparticles have capsaicin and/or proteins linked to the chalcogen atoms localized on the surface of Se- and TeNPs, which may suggest that such biomolecules may be acting as capping agents of the nanostructures. For instance, Lomelí-Rosales et al. proposed recently that amino acids found in leaf extracts of Capsicum chince (Habanero pepper) selectively coordinate with Ag nanoparticles (Ag NPs), while polyphenols and reducing sugars could be involved in synthesizing the metallic nanoparticles [42]. Previously, Li et al. concluded using FT-IR spectroscopy results that proteins detected in Capsicum annuum L. extracts were responsible for the stabilization of Ag NPs using such peppers [44].

### 3.4. TEM and SEM Characterization of the Samples

The NP size and morphology of the structures were characterized using TEM right after the samples were purified (Figure 4A–D). The JP- and HB-SeNPs were found coated with organic clusters of varying extensions. The morphology of the SeNPs was spherical and the presence of an organic coating can be noticed in the images. These organic clusters were easily pulled apart after a few seconds of sonication, leading to monodispersed nanoparticles in solution. Nanospheres were also seen for JP- and HB-TeNPs. The particle size distributions of the nanostructures are summarized in Table 1 (including the standard deviations) after TEM imaging.

Scanning electron microscopy (SEM) and energy-dispersive X-ray studies were conducted to further demonstrate the size and morphology of the NPs and elucidate their elemental composition. SEM measurements (Appendix A, Section A.2.) demonstrated that the SeNPs were generally larger than the TeNPs, and more spherical in shape, while organic debris derived from the natural extracts and employed in the synthesis process were present around all samples. Differences in shape can be explained via crystallographic orientations and phases in both elements when reaching the nanoscale and growing from small nuclei to nanoparticles.

The NP composition was further assessed using EDX (Appendix A, Section A.3), whose data showed that the NPs are in fact made of the elements present in the metal salt solution mix used for synthesis while showcasing the presence of organic molecules (such as carbon and oxygen), mainly related to the phytocomponents associated with the synthesis of the NPs. Lastly, the crystallographic properties of both nanosystems were studied using XRD analysis (Appendix A, Section A.4), revealing a hexagonal pattern for both JP-mediated NPs, while the HB-mediated ones present an amorphous structure. Moreover, the stability of the NPs was checked as well to study the potential aggregation of the NPs over time (Appendix A, Section A.5). The study revealed no significant changes in zeta potential measurements, showing that the negatively charged nature of the NPs did not change over time, hence not tending to a substantially aggregate. 

### 3.5. Testing the Antimicrobial Effect of the Nanoparticles

To assess the antibacterial activity of both Se- and TeNPs, colony-forming unit assays were performed on both Gram-positive and Gram-negative bacterial strains (Multidrug-Resistant *Escherichia coli*, MDR *Escherichia coli* (Figure 5A–D) and Methicillin-resistant *Staphylococcus aureus*, MRSA (Figure 5E–H)). These bacterial strains were exposed to an increasing concentration of the four NP systems.

Both SeNPs synthesized by JP and HB extracts showed an almost negligible inhibition when exposed to MDR *Escherichia coli* (Figure 5A,B) that persisted over the full range of concentrations. Nonetheless, a clear inhibition pattern was found when both JP- (Figure 5C) and HB-TeNPs (Figure 5D) were exposed to the same bacterial strain, with no appreciable growth or colony formation in the plates across all repetitions of the experiment, even at the lowest concentration of 25 µg/mL.

Furthermore, a similar behavior was found when MRSA was exposed to the four nanosystems: an almost negligible inhibition was seen when both JP- (Figure 5E) and HB-SeNPs (Figure 5F) were added in increasing concentrations, and a significant inhibition of bacterial growth was seen even at the lowest concentration of both JP- (Figure 5G) and HB-TeNPs (Figure 5H). 

To further assess the antibacterial behavior of the TeNPs made by both pepper extracts, a new set of experiments was designed in which two new concentrations below the previous threshold of the lowest concentration were used, i.e., 5 and 10 µg/mL (Figure 6). The objective was to observe a pattern of antimicrobial behavior at low concentrations and gather additional data points for minimum inhibitory concentrations (MIC) analysis.

The new set of experiments revealed that the impact of the TeNPs was higher in MDR *Escherichia coli* (Figure 6A,C) compared to MRSA (Figure 6B,D), in which there was some bacterial growth at the lowest concentration that decreased when an increasing concentration of TeNPs was exposed to the bacteria. 

SeNPs show a strong growth inhibition and local destruction against many microorganisms, as previously reported in the literature [40,45]. Once the NPs contact the cells, intracellular components, such as polysaccharides, are known to leak outside the membranes. The damage to cell walls and membranes has been also observed through SEM analysis and is often associated with the extracellular overproduction of ROS [41]. On the other hand, the antibacterial mechanism of TeNPs has not been completely elucidated, even though the production of ROS is considered one of the factors involved in their antibacterial characteristics. Furthermore, TeNP-induced antibacterial mechanisms have been linked to superoxide-mediated oxidative stress causing cytoplasmic thiol oxidation, inactivation of iron–sulfur center-containing enzymes, and the peroxidation of the membrane lipids, which lead to cell death [46,47]. The minimum inhibitory concentrations (MIC) were calculated (Table 2) to further quantify the antibacterial effect of the nanoparticles for the experiments that showed an antibacterial effect.

The MIC values calculated were above or lower than those found in the literature, especially for those associated with TeNPs. Nevertheless, the same trend was not found for SeNPs. Although the presented values were lower than those found in SeNPs made from ginger extracts, which showed a MIC value of 150 μg/mL [48], for the most part, they were either higher or similar to the vast majority of values reported in the literature [49]. In the case of TeNPs, the MIC values found for the present nanomaterials were extremely low compared to others found in the literature. For instance, typical MIC values of TeNPs exposed to *Staphylococcus aureus* and *Escherichia coli* were found around 250 and 500 μg/mL, respectively [29,50].

Additionally, with the aim to study the effect of the NPs on bacterial populations after various times of exposure, SEM and cell fixation analysis were completed (Appendix A, Section A.6). The study showed that all of the NPs were able to attach to the surface of the bacteria and cause disruption and formed pores within the membrane, which led to gradual deformation and bacterial death. 

### 3.6. Testing the Effect of the Nanomaterials When Exposed to Human Cells

To determine the toxicity of the JP- and HB-Se/TeNPs on mammalian cells, in vitro cytotoxicity MTS assays were performed with human dermal fibroblasts (HDF) and human melanoma (MEL) cells using nanoparticle concentrations ranging from 25 to 100 μg/mL for periods between 1 and 3 days.

For JP-SeNPs (Figure 7A), no significant cytotoxicity was found when the HDF cells were treated with nanoparticles for 24 h over all the concentration ranges, with a slight decay in cell proliferation found at 75 and 100 µg/mL, showing a similar trend as the one found for HB-SeNPs (Figure 7B). On the other hand, TeNPs showed, as expected, a more prominent cytotoxicity effect, with a clear dose-dependent inhibition found in JP-TeNPs (Figure 7C) that ended with a total inhibition of cell growth at the highest concentrations for 24 and 72 h treatments. On the other hand, HB-TeNPs (Figure 7D) did not elicit a significant cytotoxicity behavior at 24 and 72 h for all concentrations, except at the highest concentration tested, i.e., at 100 µg/mL. 

In terms of the anticancer activity, both SeNPs, made by JP (Figure 7E) and HB (Figure 7F) extracts showed clear dose-dependent cytotoxic behavior towards the cancer cells that was especially significant at 72 h of exposure, with more than 50% depletion of cancer cell growth at concentrations as low as 50 µg/mL. On the other hand, anticancer properties were also shown for TeNPs, with a significant inhibition found for JP-TeNPs (Figure 7G) at 75 and 100 µg/mL only at 72 h, and a similar effect for HB-TeNPs (Figure 7H) that extended to a lower 50 µg/mL concentration, but also at 72 h of exposure. No relevant anticancer trends were found at 24 h other than a 50% depletion of cell growth at the highest concentration. 

Considering the observed data, and with the aim to provide some more quantitative trends, IC_50_ values were calculated (Table 3). This parameter provides a numerical view of the cytocompatibility and cytotoxicity potential of the NPs when exposed to the two cell lines. 

The data presented here revealed a greater cytocompatibility profile for both Se- and TeNPs systems when exposed to non-cancerous cells compared to melanoma cells, except for JP-TeNPs, which showed a significant cytotoxic profile towards healthy cells too. The data are also in concordance with other published data around the use of biogenic SeNPs when exposed to cancerous cells. For instance, values of 41.5 μg/mL were obtained when bacterial-synthesized SeNPs were added to a MCF-7 cell line [51]; while a decrease in IC_50_ was found when plant-synthesized SeNPs were exposed longer to A549 lung cancer cell lines, i.e., from 80 μg/mL at 24 h to 40 µg/mL after 48 h of exposure [52]. Similar trends were found for TeNPs, in which their exposure triggered lower toxicity in normal cells in comparison to cancer cells in both in vitro and in vivo models [53].

The observed anticancer properties of SeNPs have been extensively reported in the literature, with the NPs generally believed to trigger tumor cell apoptosis by enhancing cellular uptake and overproduction of ROS, as well as by stimulating cancer cell autophagy [54,55]. In the same way as what happened with SeNPs, TeNPs showed an anticancer effect related to ROS production and inhibition of relevant metabolic pathways in the cells [30,33]. Nevertheless, these mechanisms are largely unknown due to the non-extensive use of these NPs as biomedical agents. Additionally, SEM and cell fixation studies were conducted with the NPs and human cells (Appendix A, Section A.6). While no significant disruption or damage was observed when the NPs were exposed to HDF cells, signs of necrosis and membrane disruption were found when the NPs were added to melanoma cells. 

### 3.7. ROS Study

Lastly, ROS analysis (Figure 8) showed an increase in ROS production when the NPs were present in the media, with a dose-dependent effect. Therefore, the contribution of ROS was related to the NP content added to the melanoma cells, which was in accordance with the dose-dependent anticancer behavior that was shown before. 

The production of these species has already been shown when cells and bacteria are exposed to Se- and TeNPs. For instance, it has been shown that the rapid and massive accumulation of SeNPs in cancer cells is associated with a significant release of ROS [56]; the oxidative stress induced by high ROS production is regarded as a very important mechanism of antibacterial activity, as it can damage DNA, cell membranes, and cellular proteins to the point of cell death [41]. By contrast, it has been reported before that ROS production is also induced by the presence of TeNPs [29], which is associated with both antibacterial and anticancer behaviors [57]. In the specific case of mammalian cell biology, the effect of these short-lived and highly reactive molecules is often associated with damage to proteins, nucleic acids, lipids, membranes, and organelles, which can lead to activation of cell death processes such as apoptosis [58] and necrosis [59]. Nonetheless, both the antimicrobial and cytotoxicity found in the nanoparticles could not be explained by ROS. Other mechanisms of cell death have been often found to simultaneously act in conjunction with ROS, namely the release of metallic/metalloid ions [60,61] and/or the physical interactions of the nanoparticles themselves with the cellular membranes, leading to disruption via collision [57]. 

## 4. Conclusions

In this work, the natural extracts of two species of peppers (Jalapeño and Habanero peppers) were used as the unique reducing and capping agents to produce Se- and TeNPs, in a reaction that was completed in less than 30 min with the assistance of microwave irradiation in an environmentally friendly, cost-effective, and straightforward fashion. The selection of Jalapeño (*Capsicum annuum*) and Habanero (*Capsicum chinese*) as both reducing and stabilizing agents in the synthesis of Se- and TeNP was based on their abundance of useful metabolites and active raw biocomponents. The nanosystems were characterized using electron microscopy and physicochemical characterization techniques to reveal their elemental content in terms of either chalcogen element and an organic-like corona surrounding the nanoparticles because of the presence of the pepper extracts, as well as a characteristic morphology and size. Once characterized, the nanosystems were studied for their antimicrobial properties, revealing a significant bacterial inhibition for the TeNP systems, with MIC values below 2 μg/mL for both combinations when exposed to Gram-negative and positive bacteria; the SeNPs hardly showed any inhibition when applied to the same bacteria. On the other hand, when exposed to human dermal fibroblast cells, all NP systems remained cytocompatible except for JP-TeNPs, which showed a strong dose-dependent inhibition that made the system incompatible with skin-associated treatments. Nonetheless, this dose-dependent cytotoxicity was found in all four systems when exposed to melanoma cells, with a mechanism of cell death associated, among other factors, with the production of reactive oxygen species, which was found to be dose-dependent in the presence of the nanoparticles. In view of the presented data, HB-TeNPs were found to be the most relevant nanosystem in terms of their biomedical properties, with a strong antimicrobial and anticancer effect and a cytocompatibility profile at concentrations up to 75 μg/mL. Overall, the use of pepper extract was presented as a suitable and eco-friendly approach for the synthesis of chalcogen nanoparticles with potential biomedical features which can be used for the topical treatment of microbial infections and the control of melanoma-associated tumors.

## Figures and Tables

**Figure 1 jfb-14-00024-f001:**
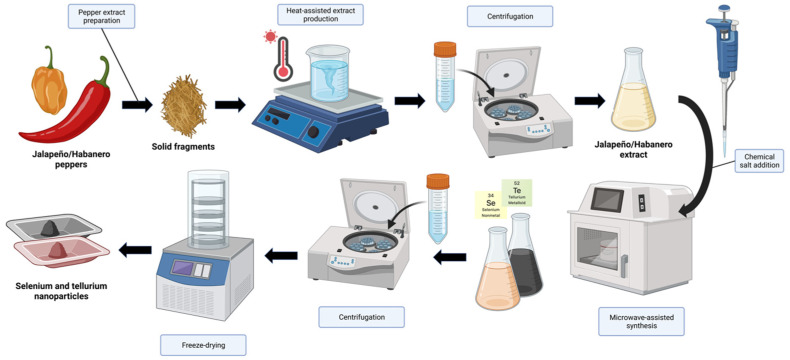
Schematic representation of the synthesis and purification of both selenium- and tellurium-based nanoparticles using pepper extracts.

**Figure 2 jfb-14-00024-f002:**
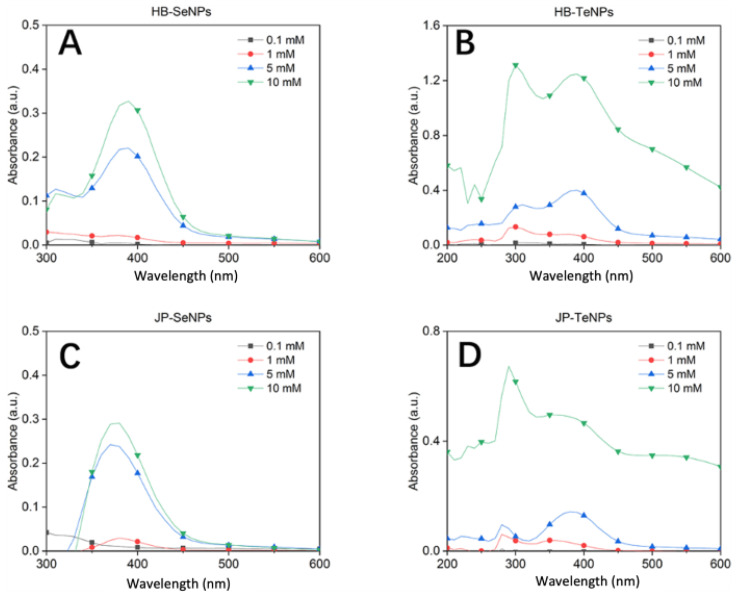
UV-visible absorbance spectra for HB-SeNPs (**A**), HB-TeNPs (**B**), JP-SeNPs (**C**) and JP-TeNPs (**D**) using different concentrations of Se and Te precursors.

**Figure 3 jfb-14-00024-f003:**
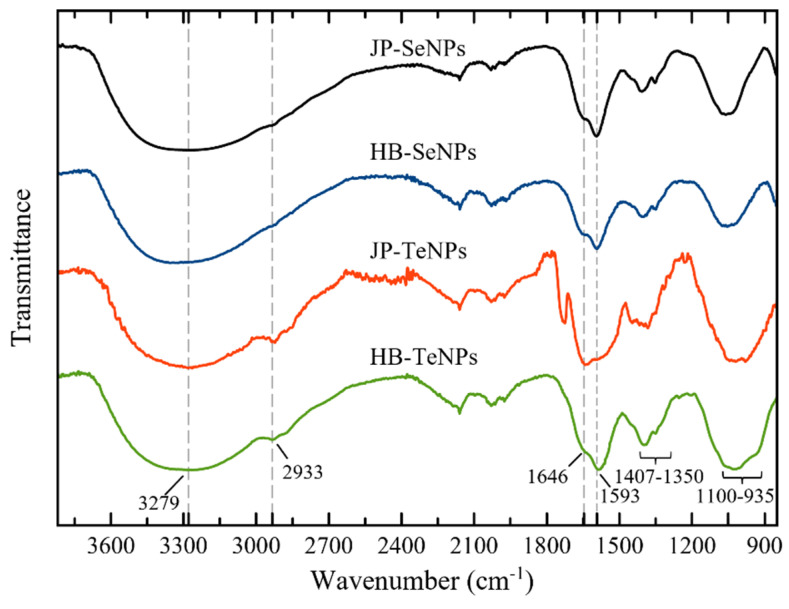
FT-IR spectra of SeNPs and TeNPs synthesized with Habanero (HB) and Jalapeño peppers (JP).

**Figure 4 jfb-14-00024-f004:**
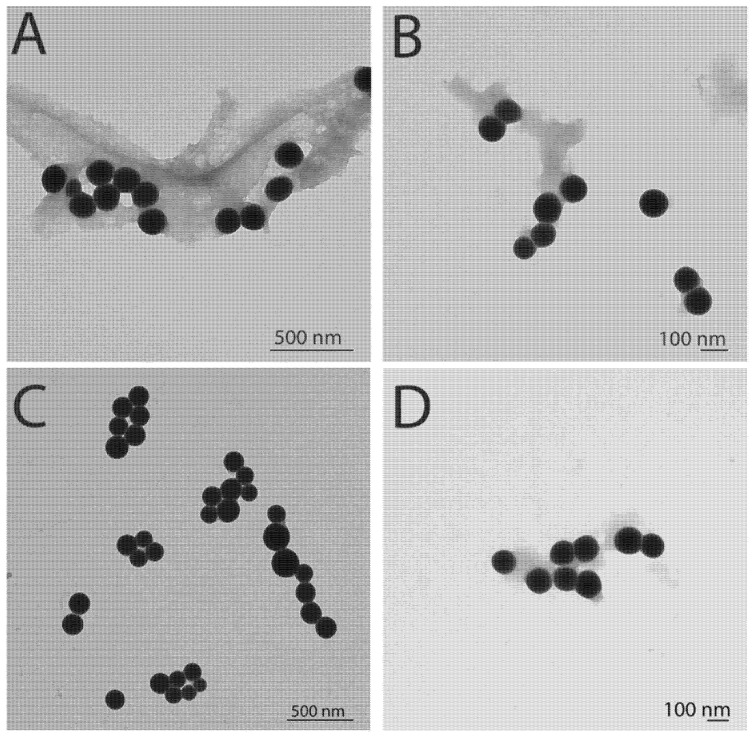
Transmission electron microscopy (TEM) characterization of the SeNPs synthesized by Jalapeño (**A**) and Habanero (**B**) extracts and TeNPs synthesized by Jalapeño (**C**) and Habanero (**D**) extracts with 10 mM of the corresponding precursor salt concentration.

**Figure 5 jfb-14-00024-f005:**
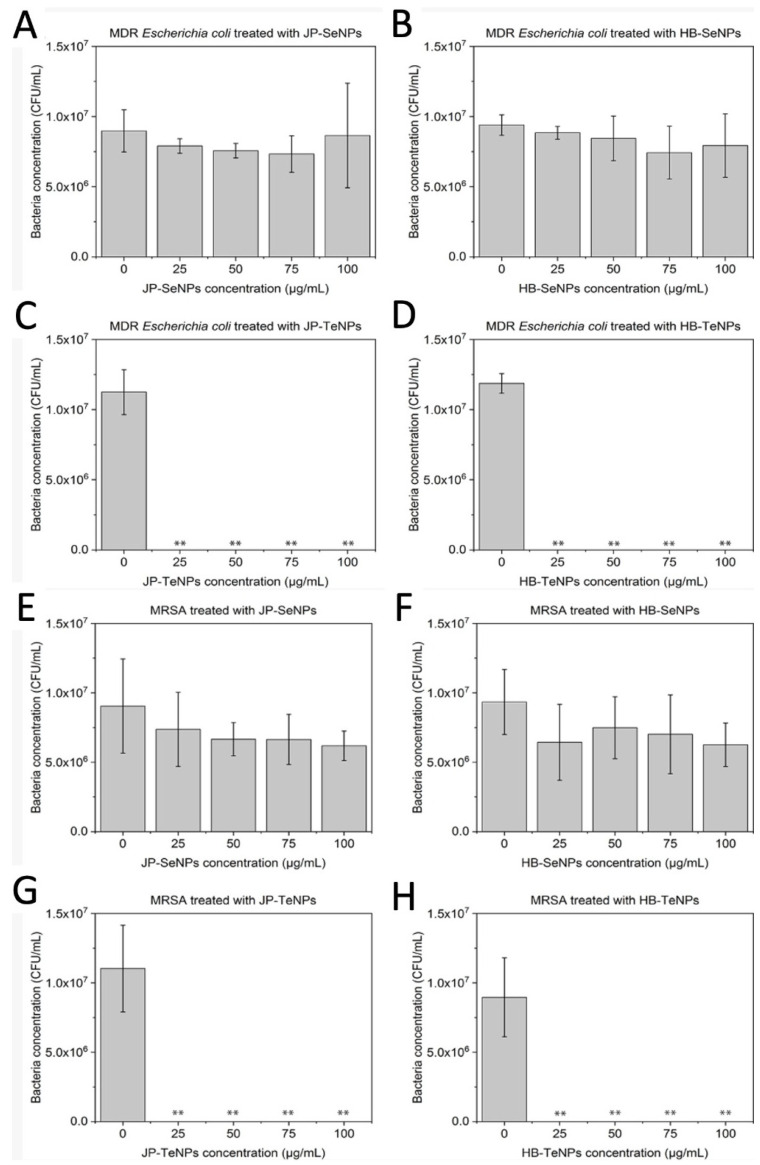
Colony counting assays of MDR *Escherichia coli* exposed to (**A**) JP-SeNPs, (**B**) HB-SeNPs, (**C**) JP-TeNPs and (**D**) HB-TeNPs and MRSA exposed to (**E**) JP-SeNPs, (**F**) HB-SeNPs, (**G**) JP-TeNPs and (**H**) HB-TeNPs. N = 3, where three measurements were taken for each biological replicate. The magnitude of the mean of the technical and biological replicates and the standard error are shown in the bar chart with error bars. ** *p* < 0.01 versus control.

**Figure 6 jfb-14-00024-f006:**
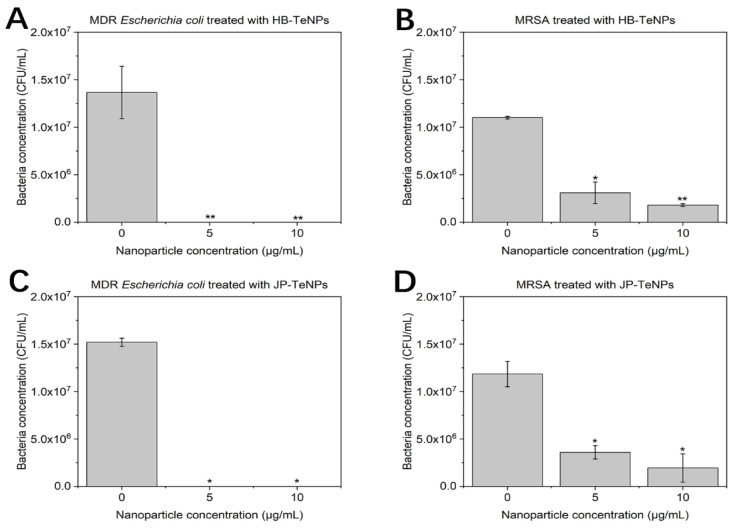
Colony counting assays of HB-TeNPs exposed to MDR *E. coli* (**A**) and MRSA (**B**), and JP-TeNPs exposed to MDR *E. coli* (**C**) and MRSA (**D**). N = 3, where three measurements were taken for each biological replicate. The magnitude of the mean of the technical and biological replicates and the standard error are shown in the bar chart with error bars. * *p* < 0.05 versus control and ** *p* < 0.01 versus control.

**Figure 7 jfb-14-00024-f007:**
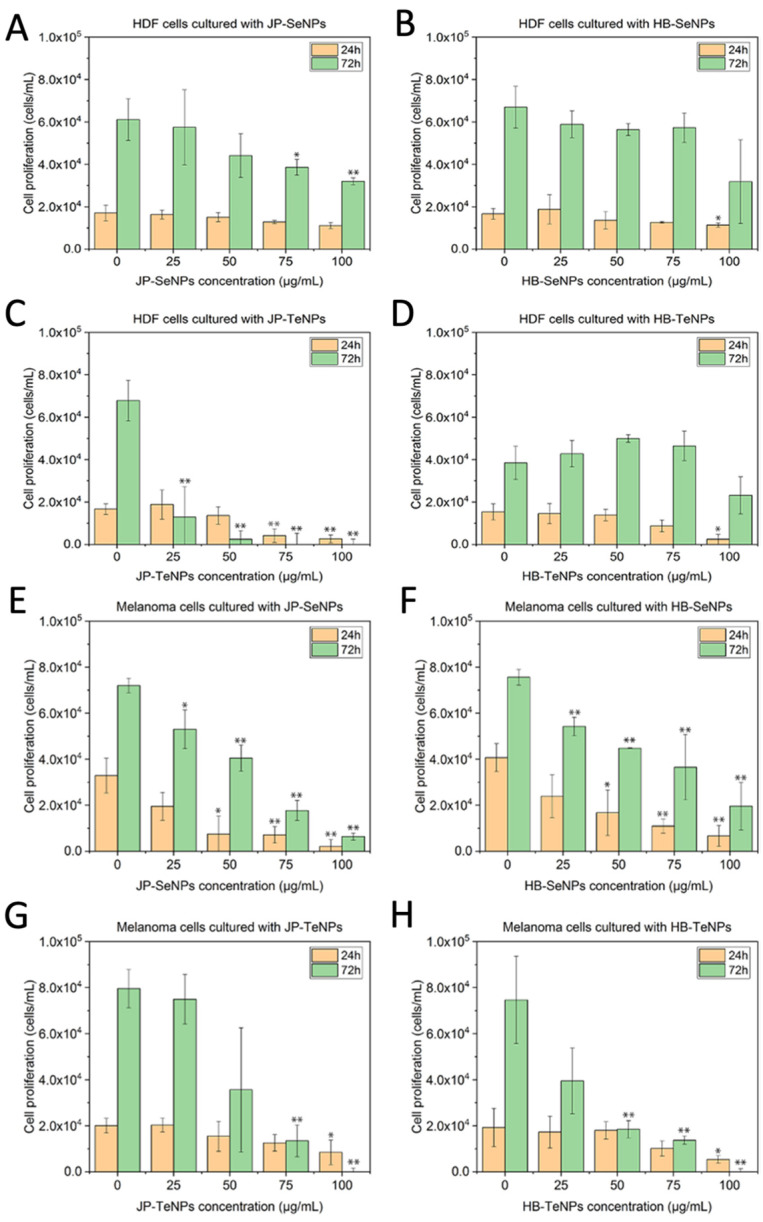
MTS cytotoxicity assay results of HDF cells exposed to JP-SeNPs (**A**), HB-SeNPs (**B**), JP-TeNPs (**C**) and HB-TeNPs (**D**) and melanoma cells exposed to JP-SeNPs (**E**), HB-SeNPs (**F**), JP-TeNPs (**G**) and HB-TeNPs (**H**). N = 3, where three measurements were taken for each biological replicate. The magnitude of the mean of the technical and biological replicates and the standard error are shown in the bar chart with error bars. * *p* < 0.05 versus control and ** *p* < 0.01 versus control.

**Figure 8 jfb-14-00024-f008:**
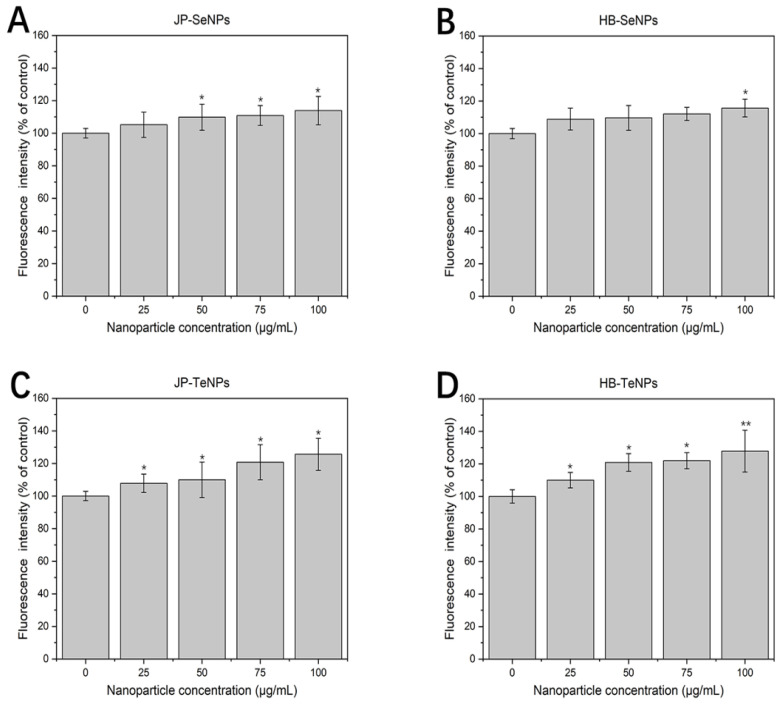
ROS measurements of MEL cells exposed to JP-SeNPs (**A**), HB-SeNPs (**B**), JP-TeNPs (**C**) and HB-TeNPs (**D**). N = 3, where 3 measurements were taken for each biological replicate. The magnitude of the mean of the technical and biological replicates and the standard error are shown in the bar chart with error bars. * *p* < 0.05 versus control and ** *p* < 0.01 versus control.

**Table 1 jfb-14-00024-t001:** Size distribution of the different NPs obtained through TEM measurements.

Nanostructure	Diameter (nm)
JP-SeNPs	79.2 ± 10.2
JP-TeNPs	82.1 ± 2.1
HB-SeNPs	90.6 ± 14.4
HB-TeNPs	74.3 ± 10.4

**Table 2 jfb-14-00024-t002:** MIC values calculated for the different NPs exposed to the bacterial strains.

MIC (µg/mL)				
Bacterial Strain	JP-SeNPs	HB-SeNPs	JP-TeNPs	HB-TeNPs
MDR *Escherichia coli*	-	72.2	0.9	1.2
MRSA	-	85.1	0.3	2.0

**Table 3 jfb-14-00024-t003:** IC_50_ (µg/mL) values calculated for the different Se- and TeNPs exposed to HDF and melanoma cells.

Cell Type	HDF Cells		MEL Cells	
Nanosystems	24 h	72 h	24 h	72 h
JP-SeNPs	74.4	52.9	28.2	49.1
HB-SeNPs	45.9	66.9	60.2	30.0
JP-TeNPs	58.6	34.1	80.8	51.0
HB-TeNPs	81.0	91.5	71.8	23.1

## Data Availability

Not applicable.

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
