# Peer review of "Pepper-Mediated Green Synthesis of Selenium and Tellurium Nanoparticles with Antibacterial and Anticancer Potential"

_jfb, 2022, doi:10.3390/jfb14010024_

Round 1

Reviewer 1 Report

This manuscript reported the synthesis of Se- and Te-NPs by using Jalapeño and habanero pepper extracts as both reducing and stabilizing agents. Then, the NPs were tested for their antibacterial properties against gram-negative and gram-positive bacterial isolates, as well as for their cytotoxic profiles with healthy dermal fibroblasts and cancerous human melanoma cells, revealing insights into the mechanisms of cell death associated to reactive oxygen species production. This manuscript is valuable for publication in Journal of Functional Biomaterials after the authors address the following concerns.

1.      Keywords, the word of “metalloids” should be deleted because tellurium is a metalloid but selenium is not. Moreover, “habanero” and “jalapeño” can also be deleted.

2.      To introduce green synthesis technology, the important relevant review (Ref.: Biomolecule-assisted green synthesis of nanostructured calcium phosphates and their biomedical applications. Chem. Soc. Rev. 2019, 48, 2698-2737) should be cited in the introduction.

3.      As Se- and TeNPs have been synthesized by using different green nanotechnology approaches, including plant extracts, so the authors need to clarify why pepper extracts are used in this work.

4.      The experimental parameters of microwave heating, such as temperature, time and power, need to be given. Moreover, what are the concentrations of pepper extracts in the synthesis of Se- and Te-NPs?

5.      The resolution in Figure 2 is too low, and the text in Figure 2 is too small to see clearly. It is suggested to separate UV-vis and TEM images into two Figures.

6.      Since the absorption intensity of UV-vis spectrum is directly related to the concentration of the sample, thus Figure 2A~D need to be tested at the same concentration of samples to be meaningful.

7.      The conclusion of “SeNPs show a strong growth inhibition and local destruction against many microorganisms, as previously been reported in literature” is inconsistent with the data in Figure 3A, B, E and F.

8.      Why is the MIC value of TeNPs in this work extremely low compared to others found in literature?

9.      As the main goal of this work is to establish the use of pepper extracts as a reproducible, environmentally friendly, and cost-effective method to produce valuable plant-mediated metalloid NPs with biomedical potential, so the advantages and functions of pepper extracts should be summarized in the conclusion.

Author Response

Dear Reviewer 1.

Thank you very much for your commenst.

We have attached our responses in PDF file.

Kind regards

The authors.

Reviewer 2 Report

The article is devoted to the green synthesis of tellurium and selenium nanoparticles. These nanoparticles are not so popular as, for example, silver or gold nanoparticles, so this work is of particular interest for the discovery of new metal nanoparticles with antibacterial and anticancer properties. It should be accepted after answering some questions:

1. The article describes the process of reducing tellurium and selenium ions to nanoparticles, including the various biomolecules participation (flavones, flavonols, thiamine, etc.). Is there any information or suggestion about bio-compounds working as capping agents because of their importance for the further biomedical application?

2. What, according to the authors, is the difference in the TeNPs and SeNPs effects on gram-positive and gram-negative microorganisms?

Author Response

Dear Reviewer 2

Thank you very much for your comments.

We have attached our responses in PDF file.

Kind regards

The authors.

Reviewer 3 Report

Comments:

In this paper, the authors have successfully prepared SeNPs and TeNPs using the extracts from fresh jalapeño and habanero peppers, and also evaluated their biomedical applications (including antimicrobial and anticancer properties against isolates of antibiotic-resistant bacterial strains and skin cancer cell lines). Biological extracts, as stabilizers and reducers, have unique advantages in the preparation of nanomaterials, providing various possibilities for the crystallization and growth of nanomaterials. However, due to the complexity of their composition, it also challenges the synthesis and mechanism analysis of nanomaterials. The authors prepared JP-TeNPs with excellent antibacterial and bactericidal activities, and HB-TeNPs with good biocompatibility and antitumor activity. I believe that it is an important work to discuss the influence of biomolecules on the morphology and composition of materials, which further influences the antibacterial and antitumor activities. There are some details listed below that the authors should address:

1. What are the main active biological components in Jalapeño and habanero extracts, and whether they are loaded onto SeNPs and TeNPs as ligands through weak chemical bonds. From the results of TEM, SEM and EDX, it can be seen that there are biomolecules from the extracts in the NPs. As the component of the NPs, whether these biomolecules participate in the process of bacteriostasis or tumor inhibition, which needs to be verified and discussed. Furthermore, the information given by the EDX results is interesting. However, the discussion is too simple. Through the proportion difference of Se, Te, Ca and other elements in the system of SeNPs and TeNPs synthesized using JP- and HB- extracts, more discussion should be given in combination with the results of bacteriostasis and tumor inhibition.

2. HB-TeNPs in 25-75 μg/ml had a good biocompatibility to HDF cells, and also had a significant inhibitory effect on tumors and bacteria, while JP-TeNPs did not show the similar biocompatibility. What causes the difference between HB-TeNPs and JP-TeNPs in biocompatibility, and a reasonable explanation should be given. In addition, TeNPs showed very good bactericidal and bacteriostatic activities. The MIC of TeNPs is lower than 2 µg/mL, which is far lower than the values reported in the literatures. The discussion on this result is also unclear.

3. Since the toxicity of nanoparticles is closely related to their chemical forms and doses, it is suggested to supplement the valence information of Se and Te through X-ray photoelectron spectroscopy (XPS) analysis.

4. The variance of some results in Figure 3-5 is too high and the results are confusing. Please confirm the accuracy of the results and give a reasonable explanation.

5. The sub-title should be “Results and Discussion” but not “Results” in Line 279. The whole part of “Results and Discussion” is not clearly described, and the logic is chaotic. It is suggested to reorganize and re-write.

Author Response

Dear Reviewer 3

Thank you very much for your comments.

We have attached our responses in PDF file.

Kind regards

The authors.

Round 2

Reviewer 1 Report

The authors of the above-referenced manuscript have adequately answered the referees' questions and carefully revised the manuscript. The current revision is a better, improved and readable version, and it can be published without any changes.